# Theoretical Evaluation of Sulfur-Based Reactions as a Model for Biological Antioxidant Defense

**DOI:** 10.3390/ijms232314515

**Published:** 2022-11-22

**Authors:** Maria Laura De Sciscio, Valeria D’Annibale, Marco D’Abramo

**Affiliations:** 1Department of Chemistry, Sapienza University of Rome, 00185 Rome, Italy; 2Department of Basic and Applied Sciences for Engineering, Sapienza University of Rome, 00185 Rome, Italy

**Keywords:** ROS, methionine oxidation, thiol/disulfide interchange, PMM

## Abstract

Sulfur-containing amino acids, Methionine (Met) and Cysteine (Cys), are very susceptible to Reactive Oxygen Species (ROS). Therefore, sulfur-based reactions regulate many biological processes, playing a key role in maintaining cellular redox homeostasis and modulating intracellular signaling cascades. In oxidative conditions, Met acts as a ROS scavenger, through Met sulfoxide formation, while thiol/disulfide interchange reactions take place between Cys residues as a response to many environmental stimuli. In this work, we apply a QM/MM theoretical–computational approach, which combines quantum–mechanical calculations with classical molecular dynamics simulations to estimate the free energy profile for the above-mentioned reactions in solution. The results obtained, in good agreement with experimental data, show the validity of our approach in modeling sulfur-based reactions, enabling us to study these mechanisms in more complex biological systems.

## 1. Introduction

In aerobic organisms, $O_{2}$ biological reduction processes lead to the formation of different by-products, such as superoxide, hydrogen peroxide, hydroxyl radical and single-oxygen, that are known and classified as Reactive Oxygen Species (ROS) [1,2]. These oxidative agents are generally able to mediate, in the cellular environment, redox signaling pathways [3,4,5,6]. Due to their role in the formation of oxygen intermediates that could interact with many cellular components (i.e., DNA, lipids and proteins) [3,7], they can be also responsible for oxidative stress and damage. Sulfur-based reactions are generally responsible for the redox signaling pathway along the cellular environment, and the residues most involved in these processes are Methionine (Met) and Cysteine (Cys), which are generally targets of ROS oxidation [8]. Cysteines act via a mechanism of disulfide bridge interchange, which implies an exchange between a deprotonated Cys in a thiolate form and an oxidizing disulfide [4,9,10], by means of a nucleophilic attack. The reaction goes through a transition state with three sulfur atoms partially linked together, sharing one negative charge on three centers [11,12,13,14]. The reversible and covalent bond formation between two sulfur atoms in a disulfide bridge in a protein environment shows similar chemical behavior to one of the small molecules. The exchange reaction has a free energy barrier of around 15 kcal/mol, with limited variations due to the polarity or hydrophobicity of the environment surrounding the reaction center [9,15,16]. The reaction of thiol/disulfide interchange is base catalyzed [17,18] and takes place in three steps [11], as shown in the following scheme:(1)$$\begin{array}{c} R-SH⇌R-S^{-}+H^{+} \end{array}$$

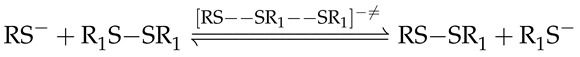
(2)
(3)$$R_{1}S^{-}+H^{+}⇌R_{1}-SH$$
where the $S_{N}2$ reaction (Equation (Equation 2)) occurs after the ionization of the first thiol (Equation (Equation 1)). In the protein context, Cys generally undergoes two-electron oxidation processes, with the formation of short-lived adducts that participate in hydrolysis. These adducts favor the formation of the S-S bridge with an additional thiol group, leading to Cystine or mixed disulfides [7]. Due to the fact that the oxidation of thiols leading to disulfides is one of the key processes against oxidative stress and ROS primary targets are represented by protein thiols, the thiol/disulfide exchange reaction plays an important biological role as an antioxidant defense.

In protein, the direct reaction can occur via a nucleophilic substitution mechanism catalyzed by specific enzymes, and it is often involved in the protein folding, (i.e., DsbA enzyme favors the formation of new disulfide bridges during the polypeptide folding process [19]). Furthermore, it is worth noting that the uncatalyzed thiol/disulfide exchange reaction is rather slow, and thus, it is usually accelerated by several orders of magnitude by the presence of specific enzymes [5], such as oxidoreductases. Thioredoxin (Trx) proteins, for example, are involved in the cellular defense system and are able to catalyze the thiol/disulfide interchange reaction due to the high nucleophilicity of the attacking Cys [5,20] (that is referred to as resolving Cysteine [4,8,21]). This behavior could be related to the active site configuration and specific relevant features, such as a low pKa of the resolving Cysteine, the presence of positively charged residues or proximity to the N-terminal part of an α-helix, where the helical dipole projects a positive density charge [4,22,23,24,25].

On the other hand, in Met residues, sulfur is found in an easy-oxidizable thioetheric form, which makes the amino acid very susceptible to ROS action. Sulfur can undergo one or two-electron oxidation based on the nature of oxidizing species [26]. The first oxidizing mechanism involves sulfide radical cation, which is a highly unstable species that can lead to fragmentation/dimerization or the formation of reactive intermediates, causing irreversible damage [7,27]. Two-electron oxidation, indeed, mainly yields to Met Sulfoxide (Met-SO) through the addition of an oxygen atom to Met sulfur, which can be easily reverted to Met, in vivo, by ubiquitous reducing enzymes, e.g., the canonical Methionine sulfoxide reductases (Msr), MsrA and MsrB [26,28]. In strong oxidative stress condition, Met-SO can be further oxidized to Met Sulfone, which is an irreversible product.




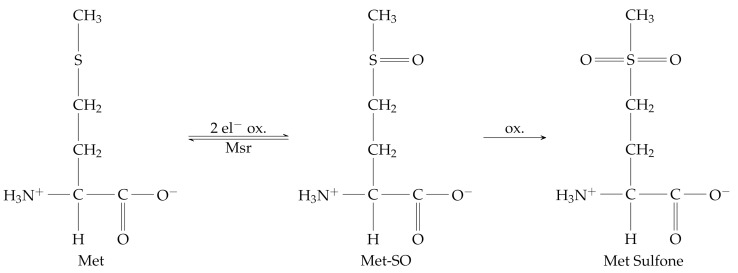

(4)



Among two-electron ROS oxidations, $H_{2}O_{2}$ oxygen transfer to Met sulfide is a clean oxidant process (i.e., the products are Met-SO and a water molecule) of high biological relevance, since hydroxide peroxide acts as a messenger—through reversible oxidations—and is itself a key modulator of redox homeostasis [29,30,31]. Under physiological condition, Met-SO is reached by the generalized accepted water-mediated mechanism of organic sulfide oxidation [27,32,33,34,35,36,37]. Experimental and computational data demonstrated the absence of acid catalysis and, consequently, of water oxide formation; a direct $H_{2}O_{2}$ proximal oxygen attack on sulfide occurs, with a simultaneous O-O/O-S bond breaking/formation, which is accompanied by a strong charge separation, and it represents the rate-determining event. Thus, water molecules surrounding the sulfur atom and $H_{2}O_{2}$ assume a fundamental charge stabilizing role and, furthermore, the post-TS hydrogen transfer, from proximal to distal oxygen, that leads to Met-SO formation, is mediated by neighboring water molecules.

Protein-bound Met oxidation can affect protein stability, structure and biological activity, due to the higher Met-SO hydrophilicity, with respect to non-oxidated Met [38,39]. Therefore, for a long time, Met oxidation has been regarded as a spontaneous protein damage process and linked to biological aging [40]. Nonetheless, in the last two decades, several efforts have been made to understand the physio-pathological role of the Met-SO/Met redox cycle. From these investigations, it has been pointed out that accumulation of Met-SO and/or dysfunction of its reducing enzymes can be linked to neurological conditions, such as Alzheimer and Parkinson diseases [27]; Met oxidation has been described as a trigger for the reversible activation of protein/part of the cell signaling mechanism (via $H_{2}O_{2}$ messenger), and it can act as a scavenger for ROS, preventing backbone oxidation and protein damage [7,41]. Hence, Met/Met-SO cycle inter-conversion plays a pivotal role in protein/cellular function reversible controlling [42,43].

To note, the oxidation and reduction of Cys and Met are complementary reactions that, in the enzymatic environment, act in concatenated reactions. In fact, Met-SO species generally require the action of Msr enzymes, whose oxidized form is reduced back again via a thiol/disulfide exchange reaction [8,28,42]—of intra/intermolecular processes (by Cys residues of Msr itself or Trx/Trx-like proteins)—to restore the catalytic site initial form. For this reason, a complete description of the mechanisms, behind the processes above discussed, has particular relevance in redox homeostasis understanding [44].

Although the biological significance of these reactions is well established, a few theoretical– computational issues dealing with the modeling of the free energy path of such sulfur-based reactions are present in the literature [13,15,33,36,45]. In particular, for Met oxidation, Chu and co-workers [33] presented a complete and accurate work, employing constrained MD and QM/MM calculations, to obtain the free energy barriers in free Met and protein environments. On the other hand, Sjoberg [36] obtained the Gibbs free energy barriers for a tripeptide containing Methione through QM calculations with a polarizable continuum model as solvent. Concerning the disulfides exchange, Bach et al. [13] and Fernandes et al. [45] used a polarizable continuum model to obtain the electronic energies along the path, while Putzu et al. [15] used a hybrid method, expanding the QM calculation to a shell of water molecules, to estimate the free energy by means of metadynamics simulations.

Here, we present the application of a different QM/MM approach, the PMM, based on the combination of classical molecular dynamics simulations and quantum–mechanical calculations, which is able to provide an accurate estimate of the thermodynamic properties of interest at a limited computational cost.

Hence, we studied, by means of an efficient theoretical–computational approach, two important chemical reactions, the thiol/disulfide exchange and the methionine oxidation in simple but still realistic environments to provide a detailed explanation of their thermodynamics and of the factors that may affect their behavior.

## 2. Results

The gas-phase properties (i.e., atomic charges, dipole moments and electronic energies) were calculated for each point of the minimum energy profiles and perturbed in the context of the PMM procedure, as reported in Materials and Methods section.

### 2.1. In Vacuum Reaction Pathways

The potential energy surface (PES) of the two oxidation processes was obtained by scanning the energy along the reaction pathway, as provided by the gas-phase potential energy surface along two coordinates defined as the difference of distances between selected atoms. These distances are: for the sulfoxide formation, S-$O_{P}$ length and $O_{P}$-$O_{D}$ bond of $H_{2}O_{2}$ (Equation (Equation 5)), while $S_{\mathrm{Nu}}$-$S_{C}$ and $S_{C}$-$S_{\mathrm{LG}}$ for the disulfide interchange reaction (Equation (Equation 6)).




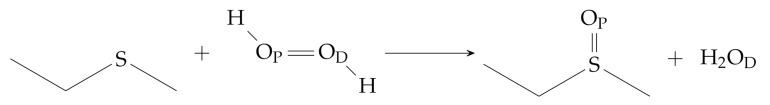

(5)






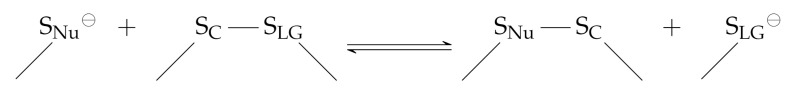

(6)



The difference between breaking and forming bond distances was used as the reaction coordinate (indicated as ξ).

${\mathrm{CH}}_{3}{\mathrm{CH}}_{2}{\mathrm{SCH}}_{3}$ oxidation PES (Figure 1, left) shows the concerted nature of the lowest energy path through which sulfoxide is formed. In fact, the transition state is located at $O_{P}$-$O_{D}$ = 2.012 Å and S-$O_{P}$ = 1.999 Å distances, ξ= 0.013 Å, meaning that the breaking and formation of bonds occur almost simultaneously. In the reactant, the hydrogen atom of the proximal $H_{2}O_{2}$ oxygen points toward sulfur electron density, as shown by Molecular Orbitals (MOs) analysis reported in Figure 2, left panel. Moving along the reaction coordinate, hydrogen shift allows the proximal oxygen and the sulfur atom electron densities to point toward each other, favoring S-$O_{P}$ bond formation. The proton transfer, from the proximal oxygen atom to the distal one, happens after the transition state and leads to the formation of a water molecule and sulfoxide. These products are significantly more stable (≈−40 kcal/mol) than reactants. The associated energy barrier of the minimum energy profile in vacuum is 28.2 kcal/mol.

For the thiol (in the anionic thiolate form)/disulfide exchange process, the selected pathway in vacuum shows a profile where reactants and products, with a negative charge localized on a single sulfur atom, are higher in energy than the transition state (Figure 1 right panel). Such a state, where the negative charge is delocalized on three centers, is arranged in a linear geometry usually found in $S_{N}2$ reactions [16,46]. Such a result is in agreement with previous studies [15,45], where a PES minimum in correspondence of the trisulfide anion—with an extended negative charge delocalization—was found. For the reaction pathway outlined above, the minimum of the gas-phase energy barrier is ≈−3.5 kcal/mol (at ξ=−0.057 Å) when the methanethiol is considered as QC.

For the description of the disulfide interchange process for the model molecules, Mercaptoethanol (ME) and Butanethiol (BU), the mercaptoethanol/2-hydroxyethil disulfide and butanethiol/dibutyl disulfide switch were described using two different QCs.

In fact, due to the relevant role played by the electronegative oxygen atoms in the molecular charge distribution, for the mercaptoethanol/2-hydroxyethil disulfide exchange reaction, the whole molecule better represents the QC, while for the description of the BU system, methanethiol/methyl disulfide were used as QCs. The in vacuum 2D and 3D PES corresponding to the Met Oxidation process and the disulfide exchange are reported in Figure 1 (right). For this last reaction, the PES scan is reported for the methanethiol/methyl disulfide system, while the ME-analogue PES scan for the disulfide bridges switch is reported in Appendix A. However, no significant differences are observed in the corresponding PESs.

### 2.2. Effect of a Perturbing Environment on the Energy Barrier

The free energy profiles for the above-mentioned processes (Equations (Equation 5) and (Equation 6)) reveal different solvent-mediated behaviors. In Met oxidation, the solvent has a strong stabilizing effect, particularly relevant in the proximity of the TS, where charge separation takes place. Noteworthy, in the perturbed profile, the TS is slightly shifted toward the reactant state (ξ=−0.752 Å). This is most likely due to an H-bond network between the solute and the neighboring water molecules that favors the S-O bond, thus stabilizing the distal oxygen atom. Therefore, the overall effect leads to a decrease in the reaction free energy barrier (14.9 kcal/mol) with respect to the gas phase, which is in good agreement with the experimental data (Table 1) [34].

Concerning the $S_{N}2$ thiol/disulfide interchange, the presence of a polar protic (i.e., water) or aprotic (i.e., dimethylformamide, DMF) solvent acts in a different way in the stabilization of reactants and products state, which is due to the ability to delocalize efficiently the net negative charge on isolated sulfur atoms. The free energy barrier for ME in water is 19.0 kcal/mol (with respect to the experimental value of 16.2 [16]), while the corresponding value associated with the BU interchange reaction is 12.2 kcal/mol, which is in rather good agreement with the experimental data (11.1 kcal/mol [16]).

In these cases, the barrier maximum of the perturbed TS is shifted toward the product states, in comparison with the in vacuum minima for methanethiol or mercaptoethanol systems (at 0.367 Å and 0.275 Å of ξ, respectively). This effect is probably related to the solvent interaction with the trisulfide anion, leading to the products.

For all the considered systems, the perturbed thermodynamic properties (i.e., Helmholtz free energy, internal energy and entropy) are reported in Table 1. The computed values of the free energy barrier are all in reasonable agreement with the experimental data reported in the literature [16,34].

The profiles of the thermodynamic properties along the reaction coordinate are shown in Appendix A.

### 2.3. Analysis of the Environment Perturbation

The analysis of the solvent effect on the perturbation of the QC was performed, within the PMM procedure, by excluding the solvent molecules from the perturbing field.

The radial distribution function (Appendix A), with respect to the sulfur atom, indicates that the first shell extends up to 5.5 Å from the sulfur atom and it is formed—on average—by 17 water molecules. In such a condition, the free energy path is similar to the complete perturbed profile, with a slight decrease of the maximum (14.8 vs. 13.2 kcal/mol), thus indicating that the water molecules in the first solvation shell well describe most of the overall perturbation (see Figure 3). In that figure, the effect of the solute atoms not included in the QC (labeled as No Water) on the free energy as well as the free energy profile of the complete system (labeled as Water) and the gas-phase potential energy are reported for comparison.

Concerning the disulfide exchange reactions, ME in water and BU in DMF both present similar unperturbed energy profiles. That is, in the gas phase, the TS region is energetically favored with respect to both reactant and product states, because of the extended delocalization of the atomic charges observed in the TS. On the other hand, the environment remarkably affects the reaction profiles, leading to free energy barriers of 19.0 kcal/mol for ME in water and 12.2 kcal/mol for BU in DMF.

## 3. Discussion

Sulfur atoms in thiolate and thioetheric form are very susceptible to ROS action. Thus, acidic Cys and Met residues play a key role in cellular redox homeostasis.

Sulfide oxidation and in particular methyl ethyl sulfide has been investigated through QM calculation, to model Met oxidation.

As described in the Introduction section, sulfide oxidation by hydrogen peroxide is a water-dependent process (i.e., charge separation stabilizers) and the need to include two or three stabilizing water molecules to reproduce experimental data was previously assessed [32,33,34,35].Our approach, based on the PMM method, allowed us to exclude selected water molecules from explicit QM calculation and, at the same time, to study the role of the solvent in the transition state stabilization. In light of such evidence, our gas-phase mechanism is, indeed, very similar to the water-mediated one, although some differences have been pointed out. In particular, in the reactant state, H of $O_{P}$ is involved in an interaction with the sulfur atom, while in the presence of water, the same atom is forming an H-bond with surrounding water molecules [33,35]. Furthermore, in the proposed mechanism, the hydrogen transfer is a direct 1,2 transfer from $O_{P}$, already bound to S, to $O_{D}$; this process was previously assessed as a 1,4 hydrogen transfer through a water molecule [33,35].

However, in both cases, this event happens after the TS and, thus, does not affect the (free) energy barrier. As reported in Figure 2 (left panel, second-last point), a tricentric HOMO, centered on S-O-O, characterizes proton transfer and products formation.

In line with the mechanism described above, the estimated free energy is in good agreement with the experimental value (Table 1 and Figure 3) when Met oxidation in solution is modeled. To note, in the reactant region (ξ=−1.8, −1.0 Å), the energy trend is independent of the perturbation due to the absence of large charge separation (points 1 and 2 of Appendix A). From this point, $O_{P}$-$O_{D}$ breaking is starting, and a charge separation occurs. $O_{D}$ becomes progressively more negative (Appendix A and Figure 2)—in agreement with previous observations [32]—and the stabilizing effect of the environment becomes more relevant.

The described process is a spontaneous reaction that can occur whenever a hydrogen peroxide molecule interacts with a Met side chain. Therefore, free Met or exposed non-burial protein-bound Met are more susceptible to oxidation and, accordingly, the number of water molecules surrounding the sulfur atom is used as a common descriptor to evaluate specific Met residue liability [35,47,48]. For this reason, a further investigation of the role of water molecules in the first coordination shell has been conducted. As reported in Figure 3, the electric field generated by water atomic charges belonging to the first solvation shell has a remarkable stabilizing effect on the Met oxidation free energy landscape. The resulting free energy profile is very similar (1 kcal/mol lower) with respect to the bulk condition.

These findings support the hypothesis of an important role played by the H-bond network, which involves water molecules surrounding the sulfur atom and $H_{2}O_{2}$. The tiny difference between Δ*A*${}^{‡}$ in the two environments is due to entropic effects, as shown in the inset of Figure 4.

Thiol/disulfide exchange (Equations (Equation 1)–(Equation 3)) is a $S_{N}2$ reaction that proceeds, along the S–S bond axis, via a linear trisulfur anionic transition state, by means of reversible reaction steps [12,13,15,45]. From the MOs analysis of the selected points along the reaction coordinate (Figure 2, right panel for methanethiol system and Appendix A for ME system), there is a clear presence, in the TS, of a high electron density on the two terminal sulfide atoms (attacking and leaving groups in the reaction) [15,49], which is represented by red surfaces in the figure. This is also observed from the ESP atomic charges (reported in Appendix A), where the points 3–4, close to the transition state maximum, show an excess of negative charge density on the terminal sulfur atoms ($S_{\mathrm{Nu}}$ and $S_{\mathrm{LG}}$), while a slightly positive charge is present on the central $S_{C}$ sulfur. Due to the symmetry of the reaction, similar ESP charges and energy surfaces in vacuum are observed for reactants and products (Figure 1, right panel). By considering the gas-phase reaction pathways, using methanethiol/methyl disulfide (for BU reaction) and mercaptoethanol/2-hydroxyethil disulfide (for ME reaction) as QCs (Figure 1), reactants and products show higher energies with respect to the TS. On the other hand, the solvent surrounding the thiol/disulfide site stabilizes the anionic thiolate species ($S_{\mathrm{Nu}}$ and $S_{\mathrm{LG}}$), resulting in an energy barrier of about 15 kcal/mol, which is due to a higher stabilization of reactants/products, with respect to the TS. According to these findings, what emerged from the experimental data of Singh and Whitesides [16] is that a polar aprotic solvent such as DMF reduces the energy barrier height—in comparison with the same reaction in water—because reactants/products are less stabilized by the environment. Therefore, if the exchange site is embedded in an environment with low dielectric constants, such as hydrophobic internal regions of proteins, the free energy barrier height decreases, thus favoring the reaction [13,15].

This solvent-stabilizing effect was reliably reproduced by the PMM-MD procedure for the two benchmark molecules used in this work, ME and BU (see Table 1). This fact is directly related to a lower stabilization of the negative monosulfur center when the environment is not polar-protic. In Figure 3 (right panel), the presence of a perturbing environment plays the same role for the two represented systems. With respect to the gas-phase profile, reactants and products, with a negative charge exposed to the solvent, are more stabilized than the trisulfide anion of the TS. The difference in the stabilization between water or DMF is mainly due to the different interaction with the sulfide anion. Polar and protic molecules of solvent (i.e., water) better stabilize the net charge, thus increasing the depth of the energy barrier with respect to a polar aprotic solvent, such as DMF. This effect, already described in the literature [16], was properly reproduced by PMM calculations, revealing a difference in the barrier maximum of 6.8 kcal/mol, in line with the experimental data, where a difference of 5.1 kcal/mol between water and DMF was measured.

It is worth noting that the positive values of $\Delta S^{‡}$ probably reflect the increase of the conformational freedom of the solvent in the TS with respect to the reactants. In fact, the interaction of (polar) solvents with a polar solute is expected to be higher when the solute has a net charge with respect to the TS, where it is spread around different atoms. This effect overcompensates the reduced conformational freedom of the TS, where from two molecules (reactant state), a single molecular adduct is formed. The internal energy and entropy profiles along the reaction coordinate are shown in Figure 4 (right) for ME in water and BU in DMF.

In conclusion, thiol/disulfide interchange and Met side chain oxidation reaction appear as two antioxidant competitive–cooperative mechanisms (i.e., share similar free energy barrier), through which a cellular redox steady state can be preserved in the presence of ROS or in other stress conditions. Our work, with the aim of providing mechanistic insights into these sulfur-based reactions, was able to model these kinds of reactions with high accuracy at a limited computational cost. The effect of the solvent to the reaction free energy profile was analyzed by considering the role of the first solvation shell and of solvents with different dielectric constants (i.e., water and DMF). Our results represent the starting point for further and detailed characterizations of such two-electron oxidation processes occurring in more complex environments, such as proteins.

## 4. Materials and Methods

### 4.1. Theoretical Background: The Perturbed Matrix Method (PMM)

The hybrid Quantum Mechanics/Molecular Mechanics (QM/MM) approach used in this work is the Perturbed Matrix Method (PMM) [50,51,52,53], that, similarly to other hybrid calculations, consists of the quantum–chemical treatment of a limited region of the system, which is defined as the Quantum Center (QC). What remains outside the QC constitutes the perturbing environment, which is described by means of computationally efficient classical Molecular Dynamics (MD) simulations. Due to its accuracy and computational efficiency, the PMM-MD is revealing a method particularly suitable for the description of processes occurring in (complex) biological systems [50,51,54,55,56,57]. In this context, it was applied for the evaluation of the free energy reaction pathways related to the oxidation of sulfur-based amino acids, i.e., the Methionine oxidation and the thiol/disulfide interchange as a model for Cysteine catalytic activity.

According to PMM procedure, the effect of the environment is included by building and diagonalizing a perturbed Hamiltonian matrix, which is represented by the electronic Hamiltonian operator, $\hat{H}$, that is composed of two contributions, one unperturbed, ${\hat{H}}^{0}$ and one given by the perturbation, $\hat{V}$, whose mathematical aspect is defined as in Equation (Equation 7).
(7)$$\hat{H}={\hat{H}}^{0}+\hat{V}$$

The ${\hat{H}}^{0}$ term is the isolated QC unperturbed Hamiltonian, while $\hat{V}$ is the perturbation operator, that can be obtained via a multipolar expansion—explicitly treated up to the dipolar term—centered in the QC center of mass, $r_{0}$:(8)$$\hat{V}≅\sum_{j}\left[V\left(r_{0}\right)-E\left(r_{0}\right)\cdot \left(r_{j}-r_{0}\right)+\cdots \right]q_{j}$$

The *j*-index refers to all the QC nuclei and electrons; $q_{j}$ is the charge for each *j*th particle and $r_{j}$ the corresponding distance of each *j*th particle from the center of mass. The V term is the electrostatic potential and E is the electric field, which is exerted by the perturbing environment.

In this study, a more recent version of the PMM approach, with higher-order terms derived from the expansion of the perturbation operator around each atom of the QC (atom-based expansion) [52], is used. Within such an approach, the perturbation operator, $\hat{V}$, is expanded within each *N*th atomic region around the corresponding atomic center $R_{N}$ (i.e., the nucleus position of the *N*th atom of the QC), as defined in the following equation:(9)$$\hat{V}≅\sum_{N}\sum_{j}\Omega_{N}\left(r_{j}\right)\left[V\left(R_{N}\right)-E\left(R_{N}\right)\cdot \left(r_{j}-R_{N}\right)+\ldots \right]q_{j}$$
with *j* running over all QC nuclei and electrons, *N* running over all QC atoms, and $\Omega_{N}$ a step function, being null outside and unity inside the *N*th atomic region. The expansion of the perturbing term is used in this work only for the Hamiltonian matrix diagonal elements, whereas the other Hamiltonian matrix elements are obtained by using the QC-based perturbation operator expansion within the dipolar approximation (Equation (Equation 8)). For each MD frame, the atomic motion, as described by all-atoms MD, furnishes the instantaneous perturbation, thus providing perturbed eigenvalues and eigenvectors, which are used for the evaluation of the QC ground state energy along the reaction coordinate.

### 4.2. Helmholtz Free Energy Change along a Reaction Coordinate

According to what was previously reported in the literature [51,58], by defining the perturbed energy change as a function of the reaction coordinate, ξ, we could consider the reaction as a sum of steps between ensembles along the reaction coordinate, which are typically described by a reactant state (R), a transition state (TS) and a final state, represented by the reaction products (P). Each ensemble constitutes a different environment (in practical terms, a different MD simulation with proper constraints used), while along the transition R→TS or TS→P, the energy change is typically evaluated by the mean energy calculated in the two corresponding ensembles (TS and R or P). From a generic transition between a point *i* and *i* + 1 along the reaction coordinate, the energy difference could be defined as follows:(10)$$\begin{array}{c} \Delta A_{i\to i+1}\left(\xi \right)=-k_{B}T\ln{\langle e^{-\beta \Delta H}\rangle }_{i\to i+1}\approx -k_{B}T\ln{\langle e^{-\beta \Delta U}\rangle }_{i\to i+1} \end{array}$$
where the approximation of ΔH, that is the QC-environment whole ground energy difference, with the QC perturbed electronic ground state energy difference, ΔU, is adopted. The same approach can be used for each point selected along the reaction pathway. The sum of each free energy variation (Equation 11), calculated between two consequential points, constitutes the overall free energy profile along the reaction coordinate.
(11)$$\Delta A_{tot}\left(\xi \right)=\sum_{i=1}^{n}\Delta A_{i\to i+1}$$

In the above equation, the sum goes from 1 to *n*, where *n* is the total number of points selected along the reaction coordinate. The same iterative approach was applied to estimate the total internal energy variation along the reaction coordinate.

Furthermore, the reasonable assumption that, for a rather rigid and small QC, the perturbed energy is a function of only the reaction coordinate (being independent by other internal coordinates of the QC) [58] enables us to calculate the QC unperturbed gas-phase properties (energies, dipoles and charges) of each reaction step, maintaining all the other internal coordinates fixed as a reference ensemble configuration.

### 4.3. Computational Details

The in vacuum unperturbed reaction pathway, for both the reactions, was defined by scanning the potential energy surface with Gaussian utility Mode Redundant, selecting a proper step, from reactants to products. The minimum energy path was selected by defining a Generalized Internal Coordinate, GIC, (i.e., the difference between bond distances, d($O_{P}-O_{D}$)-d($O_{P}$-S) for the sulfoxide formation and d($S_{C}-S_{\mathrm{LG}}$)-d($S_{C}-S_{\mathrm{Nu}}$) for the thiol/disulfide interchange reaction. The number of selected points (*n*) is equal to 25 for sulfide oxidation and 30 for disulfide interchange reaction.All quantum–chemical calculations were performed using Gaussian 16 [59]. The unperturbed properties (i.e., electronic energies, dipoles and ESP charges) for the first three excited states were evaluated by the means of DFT/TDDFT. In particular, the B3LYP functional was used while 6-31G(d), 6-31+G(d) and 6-311++G(2d,2p) were tested as basis sets, for both systems, to evaluate the best compromise between efficiency and computational cost. Despite the suitability of 6-31G* to describe the electronic properties of small systems [60], the basis set with diffuse functions was preferred and used in this work [13].

For this work, 6-311++G(2d,2p) was chosen to study ethyl methyl sulfide oxidation, in order to keep the same basis set split-valence and diffuse functions, which is regarded as mainly responsible for energy differences within this kind of reaction [32], as well as to maintain a good computational efficiency. Following the results of Chu and co-workers [32], the 6-311++G(2d,2p) basis set was selected, because it achieves the best compromise between accuracy and computational costs.

For the modelling of thiol/disulfide exchange, two benchmark systems were investigated in different solvents (water and DMF): (i) one consisting a molecule of a deprotonated mercaptoethanol (ME) and a 2-hydroxyethil disulfide molecule and (ii) one formed by butanethiol (BU) and its disulfide counterpart (dibutyl disulfide). For the geometry optimization, the B3LYP/6-31G* functional was used [45], while B3LYP/6-311++G(2d,2p) was used for the calculation of unperturbed properties [13].

The corresponding energy profiles, for both thioether oxidation and disulfides exchange processes, using the three different basis sets employed in this work, are reported in Appendix A. For Met oxidation, the QC is composed by ${\mathrm{CH}}_{3}{\mathrm{CH}}_{2}{\mathrm{SCH}}_{3}$ and $H_{2}O_{2}$, while in thiol/disulfide interchange, methanethiol/methyl disulfide were selected as the reactive subparts for Butanethiol. For the description of the reaction involving ME (thiolate and disulfide species), the entire molecules were chosen as QCs, because sensitivity tests on smaller QCs indicated that they are not able to reproduce the charge distribution of the whole molecule. For Met and BU systems, hydrogen was used as the link atom (for both ethyl methyl sulfide and methanethiol molecules) and treated as part of QCs.

The MD simulations, lasting 150 ns, needed to apply the PMM-MD procedure were carried out by means of Gromacs 2019 software package [61,62], keeping the temperature constant (300 K), by means of the velocity rescaling algorithm [63]. Long-range interactions were estimated using the particle mesh Ewald method, while a Verlet cut-off scheme was employed; a time step of 2 fs was used and the trajectory snapshots were recorded every 2 ps. Solvent molecules (water in the case of Met and ME and DMF in the case of BU) were added to a cubic simulation box larger enough to avoid boundary effects. The CHARMM36 forcefield was used for Met oxidation, because topology parameters of $H_{2}O_{2}$ and Met-SO were both included in this force field. For solute atoms included in the QC, the ESP charges as provided by DFT calculations were used. For ME and BU MD simulations, the topologies of the solutes and of the DMF were generated by means of the acpype software [64], while for the description of water molecules, the TIP3P model was chosen. The statistical errors associated to $\Delta A_{calc}^{‡}$, $\Delta U_{calc}^{‡}$ and T$\Delta S_{calc}^{‡}$ were estimated by means of the block-averaging procedure.

## Figures and Tables

**Figure 1 ijms-23-14515-f001:**
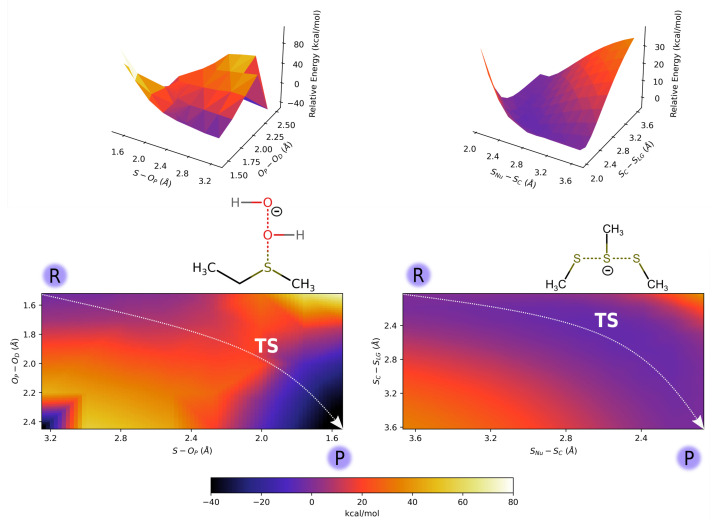
In vacuum 2D and 3D energy scan for ethyl methyl sulfide oxidation (**left**) and thiol/disulfide interchange reaction (**right**). The TS structures of corresponding QCs are reported in figure.

**Figure 2 ijms-23-14515-f002:**
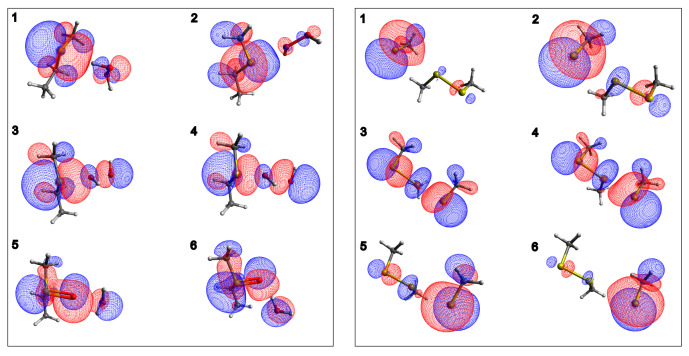
QC Highest Occupied Molecular Orbitals for six selected points of (**left**) Met Oxidation and (**right**) BU thiol/disulfide interchange reactions. In both cases, the MOs shape and direction underline the bond breaking/formation between interacting species. In the graphical representation, C atoms are in gray, O in red, S in yellow and H in white while blue (red) color highlights positive (negative) electron density. ESP atomic charges are reported in Appendix A.

**Figure 3 ijms-23-14515-f003:**
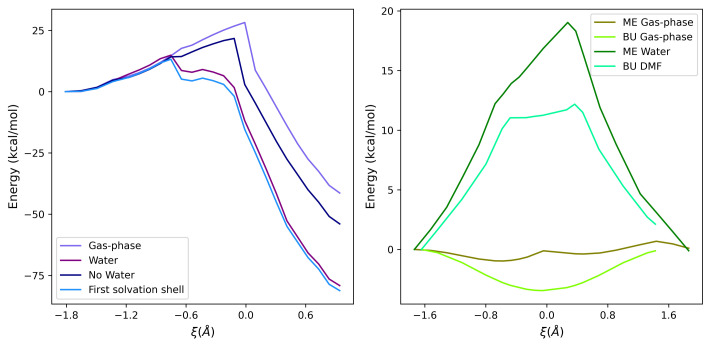
Effect of the environment on the energy profile for (**left**) Met oxidation and (**right**) thiol/disulfide interchange. All the curves, except the gas-phase one, refer to the calculated free energy.

**Figure 4 ijms-23-14515-f004:**
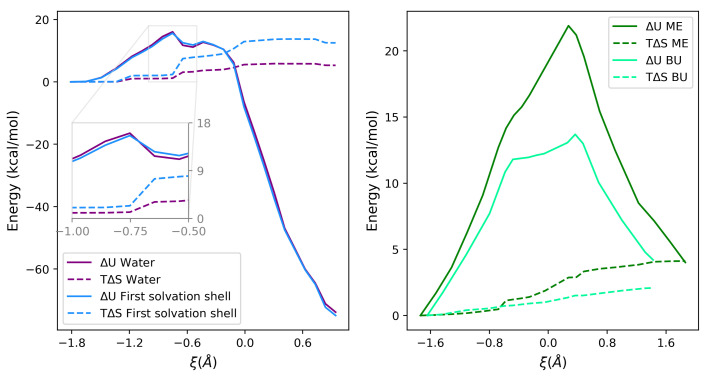
Comparison of the internal energy and entropy variations in different perturbing environments. (**Left panel**: Met oxidation; **Right panel**: thiol/disulfide interchange reaction for ME (water) and BU (DMF)).

**Table 1 ijms-23-14515-t001:** Comparison between experimental and theoretical data of the thermodynamic properties (in kcal/mol) for the Met oxidation [34] and thiolate/disulfide interchange reaction for mercaptoethanol (${\mathrm{OHCH}}_{2}{\mathrm{CH}}_{2}\mathrm{SH}$) and butanethiol (${\mathrm{CH}}_{3}{\mathrm{CH}}_{2}{\mathrm{CH}}_{2}{\mathrm{CH}}_{2}\mathrm{SH}$) [16]. The estimated statistical errors for $\Delta A_{calc}^{‡}$ and $\Delta U_{calc}^{‡}$ are 0.2 kcal/mol, while for $T\Delta S_{calc}^{‡}$, it is 0.4 kcal/mol.

Molecule	Solvent	$\Delta G_{\exp}^{‡}$	$\Delta A_{\mathrm{calc}}^{‡}$	$\Delta U_{\mathrm{calc}}^{‡}$	$T\Delta S_{\mathrm{calc}}^{‡}$
Methionine	$H_{2}O$	15.5 ${}^{a}$	14.9	16.1	1.2
Mercaptoethanol	$H_{2}O$	16.2 ${}^{b}$	19.0	21.9	2.9
Butanethiol	DMF	11.1 ${}^{b}$	12.2	13.7	1.5

*a* Experimental data reported in [34]; *b* experimental data reported in [16].

## Data Availability

Not applicable.

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
