# Peer review of "Theoretical Evaluation of Sulfur-Based Reactions as a Model for Biological Antioxidant Defense"

_ijms, 2022, doi:10.3390/ijms232314515_

Round 1
Reviewer 1 Report
The article written by De Sciscio et al., is QM/MM theoretical-computational approach, which is used to obtained results for sulfur-based reactions as a model for biological antioxidant defense. The adopted approach in modelling sulfur-based reactions is used to study the above mechanisms. Nonetheless, there is still adequate scope for improving the result analysis by explaining the trends observed with convincing explanations/ arguments.
1. The introduction is needed to be more specific about problem statement and authors must include reference work with similar computational studies.
2. The Perturbed Matrix Method (PMM) which uses truncation of the perturbed Hamiltonian matrix is how different & reliable against ONIOM technique as implemented in Gaussian 16?
3. In MD, the solvation parameters need more clarity
4. In Figure 2. The Highest Occupied Molecular Orbitals only make sense if one can see atomic symbol underneath it
5. In Figure 3, what is meaning of water and no water?
6. Please provide a worksheet in SI for calculation of ∆Acalc‡ and ∆Ucalc as given in Table 1 along with all energy units. It not sure how authors reach to final values?
Reviewer 2 Report
The manuscript by D’Abramo and coworkers deals with the theoretical calculation of the free energy of two simple reactions involving sulfur atoms. They applied a mixed quantum mechanical/classical mechanical approach to two model systems to illustrate the methodology and its power, the sulfoxide formation by hydrogen peroxide and the thiol/disulfide interchange of S. These two systems are small enough and simple for an illustration of the method, the comparison of energies with experimental data and the discussion of the role of solvent.
The reactions are quite important in biology, involving cysteines and methionine in presence of reactive oxygen species. There are many contribution in literature about the role of ROS on the biology of cells, in particular on stress condition and signaling.
The manuscript is well organized and written, I have only a few concerns before its publication. Because they want to extend in future the method to more realistic biological systems, namely the same reactions in proteins, they should discuss the two following points:
· In the end of 90 the group of Rotshilberger proposed another QM/MM scheme tailored for use in biological systems (Vande Vondele, Laio and Rotshilberger), the authors should comment and compare their method to that one
· One of the main problem is to establish q=what is quantum and what is classic. Since the simplicity of the model used here, there was not atom on the frontiers. In case of a most large model system, how do you treat the atoms at the frontiers between QM and MM groups?
They should also comment on the performance on a CPU or GPU platform to have an idea how can be extended to larger systems, please provide some benchmarks.
